# A Random Forest Model for Travel Mode Identification Based on Mobile Phone Signaling Data

**Zhenbo Lu \*, Zhen Long, Jingxin Xia and Chengchuan An**

Intelligent Transportation System Research Center, Southeast University, Nanjing 210000, China; 220173150@seu.edu.cn (Z.L.); xiajingxin@seu.edu.cn (J.X.); anchengchuan@seu.edu.cn (C.A.)

\* Correspondence: 101010360@seu.edu.cn; Tel.: +86-1367-518-9000

**Abstract:** Identifying and detecting the travel mode and pattern of individual travelers is an important problem in transportation planning and policy making. Mobile-phone Signaling Data (MSD) have numerous advantages, including wide coverage and low acquisition cost, data stability and reliability, and strong real-time performance. However, due to their noisy and temporally irregular nature, extracting mobility information such as transport modes from these data is particularly challenging. This paper establishes a travel mode identification model based on the MSD combined with residents' travel survey data, Geographic Information System (GIS) data, and navigation data. Using the data obtained from Kunshan, China in 2017, enriched with variables on the travel mode identification, the model achieved a high accuracy of 90%. The accuracy is satisfactory for all of the transport modes other than buses. Furthermore, among the explanatory variables such as the built environment factors (e.g., the coverage rate of a bus stop) are in general more significant, in contrast with other attributes. This indicates that the land use functions are more influential on the travel mode selection as well as the level of travel demand.

**Keywords:** Travel mode identification; Random forest; Mobile-phone signaling data; Residents' travel survey data

---

## 1. Introduction

Traffic demand forecasting is well recognized to be a cornerstone of urban traffic planning. Predicting urban traffic demand is of great significance for the proper management and control of urban transport systems. Existing methods for traffic demand forecasting (e.g., traffic assignment models) heavily rely on the estimation of the travel demand in different travel modes. However, it is not a straightforward task to accurately predict the demand in a multimodal transport system, due to its highly complex nature. Herein, a key hurdle is that the operation of transport system and the temporal/spatial fluctuations of traffic flows are not detectable. In the era of big data, however, mobile-phone Signaling Data (MSD) has provided a good aid for the dynamic detection of traffic flows in the entire multimodal transport system. We particularly focus on travel mode identification, mainly due to the following two reasons: (1) Understanding the travel modes people take is the key to travel behavior studies [1]; (2) The process of travel mode detection often involves data cleaning, segmentation, and inference, which are common to many motilities and urban planning applications. Some recent studies on the analysis of MSD only focus on the broad spectrum of their applications [2] (including social network analysis, mobility analysis, event detection, and urban planning), and the topic addressed in this paper is not thoroughly discussed. This paper thus focuses on identifying the travel demand of each mobile-phone user based on multiple types of data, which is a strong basis for urban travel demand forecasting.

## 1.1. Literature Review

When it comes to the research of travel mode identification, there seems to be a preference for using rule-based methods [3,4] and fuzzy logic systems [5]. Some studies also use supervised learning methods such as random forests. Rule-based methods extract the features in specific data sources and identify different modes by setting a corresponding threshold. However, this method struggles with low generalizability as rules obtained from one city may not be so applicable to another city due to various reasons. In addition, the simplicity and comprehensibility of this method mean that the derived parameters for each city must be feasible (length or average distance for each travel, for example). Fuzzy logic (FL) systems are powerful predictive models as they can handle uncertainty and vagueness in a way that is understandable by humans. However, FL is mostly subjective because the success of a fuzzy expert system lies in proper selection of its functions and parameters, which is usually done manually [1]. And the other drawback of FL systems is that these rules tend not to take into account inter-variable correlation, and as the rules are generated using experts' understanding of the field, any class (mode) additions to the mode would be extremely costly. So due to certain limitations for setting rules and algorithms in programs, some studies have turned to machine learning methods instead. And random forest is one of the most commonly used methods. Bolbol et al. (2012) estimated travel modes with 88% accuracy using the Support Vector Machine (SVM) classification methods [6,7]. Jahangiri et al. (2015) carried out a comparative study using different supervised learning methods from the field of machine learning to develop multiclass classifiers [8]. The study concluded that RF (Random Forest) and SVM methods produced the best performance. Bantis et al. (2017) used dynamic Bayesian Networks to explore the impact of incorporating both environmental and individual characteristics of travelers in the task of travel mode detection [9].

A crucial factor in the accuracy of machine learning models is feature extraction, which determines the upper-bound of their performance [10]. Zhou (2017) used the velocity statistical feature quantity, acceleration statistical feature quantity, direction change rate, and travel feature quantity to distinguish each travel mode [3]. Silanowicka et al. (2016) used various characteristic parameters like the average speed, average distance, frequency, gender, and age, in combination with background spatial information of traffic routes to differentiate travel modes [11]. Despite most features used in travel mode identification studies being limited to distance, time, speed, etc., the spatial and temporal characteristics of the trajectory of various travel modes are often associated with disturbances, resulting in reduced recognition accuracy.

Another essential consideration for travel mode identification is the data quality. For instance, the traditional four-step approach (Trip Generation, Trip Distribution, Mode Choice, and Route Choice) is widely used in the existing software packages/tools for transport system analysis and flow predictions [12]; the four-step approach highly relies on the residents' survey data. However, due to the high cost, small sample size, and long update cycle of collecting survey data, the accuracy of the four-step approach is inherently undermined by the low data quality. In view of the limitation, using high-quality automatically collected big data is a new trend in travel mode identification. Feng and Timmermans (2016) detected travel modes and activities using 142 days of GPS data from 32 volunteers [13]. In their method, four travel modes were classified based on empirical values of median speed, average speed, maximum speed, and standard deviation of speed. This method based on GPS data offers apparent advantages over traditional methods, as it requires less effort from the respondents, provides greater spatial and temporal precision and detail, and reduces the labor and time costs for the researcher [14].

However, GPS-based methods commonly rely on data from a small number of volunteers and are, therefore, not suitable for estimating mode shares on a large-scale [15]. Apart from the GPS data, Hu and Song (2015) analyzed personal travel behavior using features from smartphone sensors [16]. Though this kind of data can provide accurate speed and acceleration at a low cost, the feature extraction is difficult to perform [17,18]. With the rapid development in telecommunication networks, MSD has been recently used in mode identification because it can produce a huge amount of information

regarding how people (with their mobile devices) move and behave over space and time. While GPS data and smartphone sensors data, typically collected by smartphone apps, are restricted to rather small samples of the population, MSD are routinely collected by mobile network operators. It has full temporal coverage with comparatively low costs, potentially allowing us to analyze the travel behaviors and social interactions of the whole population. Therefore, recent years have seen an increasing interest in using such data for human mobility studies [19]. The recognized travel modes of users can then be used in different ways, such as improving traffic survey methods, developing safety applications, providing traveler information, etc. Table 1 below summarizes the advantages and disadvantages of these four data sets.

**Table 1.** Data Source Introduction.

| Data Sets | Advantages | Limitations |
| --- | --- | --- |
| Residents' survey data | Get accurate travel mode and mature use method | Time-consuming, labor-intensive and exist many human factors in the acquisition process |
| GPS data | Get a travel trajectory, real-time and less affected by external influences | Not available or may be lost in some areas and need volunteers |
| Smartphone sensors data | Get accurate speed and acceleration and less energy consumption | Some sport patterns not significantly different and feature extraction more complicated |
| Mobile phone signaling data | Full sample, real-time and match personal information | Inaccurate spatial position and difficult to extract information |

*1.2. Objectives and Contributions*

This study aims to establish a travel mode identification model based on mobile signaling data, combined with residents' travel survey data, Geographic Information System (GIS) data, and navigation data. Though systematic reviews of studies on travel mode identification highlight that machine learning methods are widely applied to this topic, they are unable to provide recognition accuracy in complex environments because the selected features are limited to spatial and temporal characteristics like distance, time, and speed. Recently, the application of MSD in solving traffic planning problem has been gaining popularity among researchers, but its use in mode identification is still limited. How to extract rich features and get tagged data are some of the difficulties that persist [20,21]. To sum up, the contributions of this paper are threefold: (1) A travel path map matching method based on base station and road network intersection is proposed to obtain the spatial-temporal characteristics of the trajectory; (2) A travel mode label sample acquisition method based on the residents' travel survey data is proposed. The travel labels of partial mobile phone track samples can be obtained by matching the residents' travel survey data with travel track data of the mobile phone; (3) Multivariate data like MSD, residents' travel survey data, and data collected from AMAP, which is a web map, navigation, and location-based services provider in China, are used to extract features. The study takes into account personal attributes and environmental characteristics of traffic facilities that are seldom used in the existing studies, while combining the navigation information to improve the performance of the proposed model in the complex traffic systems.

The remainder of this paper is organized as follows: Section 2 introduces the methodology, including map matching based on MSD, the sample label acquisition method based on the residents' travel survey data, feature extraction and selection, and the model establishment and evaluation. Section 3 presents the difficulties in data processing and application used in the paper while considering Kunshan City, China as an example location to introduce data characteristics. Section 4 conducts a case study to implement a random forest model for travel model identification and analyze the identification results. Finally, Section 5 concludes this study and suggest future research directions.

## 2. Methodology

### 2.1. Problem Description

The main objective of pattern recognition is to assure that the classification results are consistent with reality, while keeping the error rate as low as possible. Generally, the basic framework of pattern recognition can be divided into five parts, namely the acquisition of original data, data pre-processing, feature extraction and selection, classifier design, and the final classification. Apart from data, feature extraction and classifier design are also important factors affecting the recognition accuracy.

Feature extraction is based on data pre-processing. Through the analysis of the characteristics, the features that directly or indirectly exhibit similarity and difference between categories are extracted. The extraction of features will directly affect the accuracy of the classification results.

In existing literature, many scholars have extracted various features for travel mode identification. Stenneth et al. (2011) used average speed, average distance, frequency, gender, and age as characteristic parameters to classify the travel mode. Also, background information of traffic routes and travel information of travelers are included in their model [22]. Wang and Chen (2018) used speed, acceleration, direction change rate, and individual traveler characteristics (age and disability) as input features combined with GIS information [23]. In the study of Dabiri (2018), the speed, acceleration, speed curve smoothness, and direction change rate were used to identify the mode of travel, and the GPS trajectory data collected by the test was used to test and establish different classifications [24–26]. In summary, at present, scholars often only consider the difference among travel modes in the temporal and spatial distribution of the trajectory classification problem. Generally, the selected features are limited to distance, time consumption, speed, etc. But in a complex traffic environment, under the influence of the different types of travel modes, the difference characteristics reflected in the travel trajectory will be disturbed. Further, there will be a crossover phenomenon, resulting in lower recognition accuracy. Therefore, finding new features from the data other than feature information from trajectory is the key to improving the feature extraction.

Classifier design is the rule designed for classifying objects. Traditional methods for mode identification often use statistical regression, such as linear regression, a polynomial logit model [27], etc. However, these models have unrealistic assumptions and require a predefined relationship between the dependent variable and the explanatory variables. Compared with statistical models, machine-learning-based classifiers can be more effectively in mode identification. Its effectiveness has been confirmed in many studies involving decision trees, neural networks, and support vector machines [14]. However, many errors and uncertainty exist in the case of MSD, which may lead to unstable performance of the model. Random forests are found to overcome these issues and have shown good learning ability to help solve predictions and classification problems. Thus, the random forest is chosen in the study as the classifier to identify the travel mode.

### 2.2. Point-to-Point Map Matching Based on Probability Statistics

From the perspective of travel mode identification, it is very important for the mining of travel mode characteristics to map the user base station trajectory sequence onto the road network to obtain the real travel trajectory. Through path matching, features can be extracted from a more accurate trajectory when the real travel trajectory on the road network is obtained. From the perspective of mode identification application, we match the user's travel trajectory to the road network, so that more accurate traffic flow information can be obtained. With this information, we are able to achieve traffic condition identification and real-time traffic flow control. Therefore, this paper studies the map matching technology based on the location information of the mobile phone base station, which lays a foundation for feature mining in traffic pattern recognition.

Existing algorithms in map matching mainly include three categories: geometry, topology, and probability statistics. In addition, the map matching algorithm based on geometric analysis can be divided into point-to-point, point-to-line, and line-to-line matching. Point-to-point is a simple search

algorithm that matches points to the nearest node on the road network. Point-to-line matches points to the nearest road section. The line-to-line matching algorithm connects the trajectory points as curves and then searches for the most similar road segments in the road network to the matched road segments. The matching algorithm based on topology analysis is based on the topological relationship of the road network, simultaneously considering the direction, speed, and road network connectivity [28]. It then matches historical information with the actual road network information.

The main principle of the probability-based algorithm is to define a possible matching area as the center of the matching point according to the positioning accuracy of the track points, take the road section within the area as the possible matching road section and calculate its matching probability to determine the best matching road. In addition to the above map-matching algorithms, matching algorithms using complex mathematical theories have emerged in recent years, such as the Bayesian inference matching algorithm, Kalman filtering matching algorithm, fuzzy logic matching algorithm, and matching algorithm based on convex optimization [29–32].

Although a range of research on map matching already exists, the MSD was not commonly used. This may be attributed to two reasons. Firstly, the algorithm involves a large amount of computation, especially in an area with a complex road network. Moreover, as the number of segments waiting for matching is large, the algorithm efficiency can be extremely low. Secondly, the accuracy of the algorithm depends mainly on the location frequency of signaling data. In this paper, to overcome the shortcomings of the path matching method using the existing map matching technology based on mobile base stations, a novel matching method based on base stations and intersections is proposed.

Based on the map-matching algorithm and the characteristics of positioning data, this paper proposes a road network matching algorithm suitable for base station positioning trajectory. The proposed algorithm allows for the spatial distance, coverage road frequency, and road network connectivity.

After preparing the necessary information records, the sequence of the travel base station track is numbered. Suppose a track is expressed as set $S_i = \{j_1, j_2, \ldots, j_n\}$, where $j_i$ represents the base station occupied in the trajectory. The following algorithm flow is then performed for each trajectory. The procedures for map matching is shown in Algorithm 1:

---

***Algorithm 1.*** *Procedures for map matching.*

---

**Step 1:** Add the road covered by each base station $j_i$ in track $S_i$ according to the base station and road information correspondence table. Obtain the frequency table of road coverage in the trajectory $F_i$.

**Step 2:** For each base station $j_i$, the intersections set $R_i$ in coverage area can be found according to the base station and interaction information correspondence table.

**Step 3:** According to the frequency table $F_i$ obtained in Step 1, mark the road $R_{max(F_i)}$ with the highest frequency in Ri and select the intersection composed of this road. Delete the base station $j_i$, if $R_{max(F_i)} = 1$.

**Step 4:** Calculate the distance D between the intersections screened from the previous step and the base station while reserving the intersection closest to the base station according to the principle of minimum distance, i.e., the intersection corresponding to the base station on the road network.

**Step 5:** Cyclic match all base stations in the trajectory sequence to obtain an intersection sequence $C_i$. For non-adjacent intersections, the shortest circuit function is used to connect them to obtain a fully connected intersection sequence on the road network.

**Step 6:** Judge whether $C_{(i-1)}$, $C_i$, and $C_{(i+1)}$, three adjacent data in the intersection sequence $C_i$ are A-B-A type data. If yes, delete the intersection $C_i$ and $C_{(i+1)}$.

---

The path matching in the study was carried out using ArcGIS and Python. The fundamental geographic information data is stored in ArcGIS in the form of shapefile, while map matching is achieved using the NetworkX module in Python. The Dijkstra shortest path algorithm is used for the connection between the intersections. The advantages for choosing this method include high efficiency, high speed, and strong applicability in complex networks [33]. The detailed flow chart is provided as follows in Figure 1.

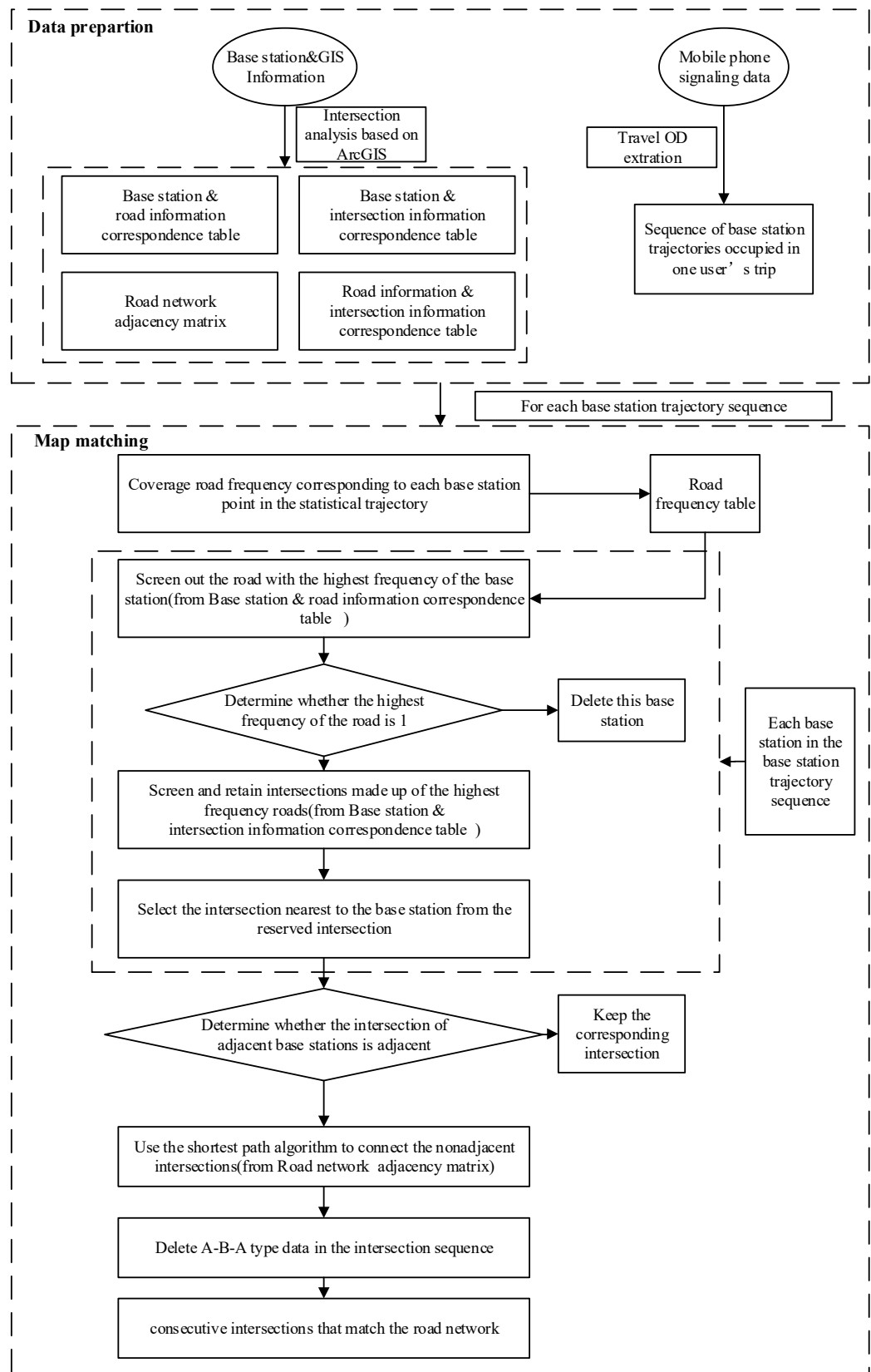

**Figure 1.** Flowchart for path matching.

*2.3. Sample Label Acquisition Based on Association Rules Mining*

MSD is location data processed anonymously by mobile operators. On the one hand, it has been widely used in many research fields due to its advantages of protecting user privacy. On the other hand, anonymity also provides few obstacles to the research. In the research of individual trip chaining with fine granularity, it is difficult to obtain the user's real travel information. Thus, it is impossible to create a link between the data and the actual scene. Further, it is difficult to verify the accuracy of scientific research results.

In previous studies, the sample data were collected by volunteers to record daily travel information. Meanwhile, the MSD was extracted from the communication operator if the volunteer consented to the data acquisition and signed the authorization letter. However, the economic cost of this method is relatively high and requires a large workforce to process the data, which is a fundamental reason for the small sample size in previous studies. Besides, due to the size and nature of the research institution, there may be a bias in the recruitment of volunteers and data collection, leading to the deviation/errors in the travel samples obtained. For example, Shafique et al. (2015) found that most volunteers for research institution projects in universities are students, whose activity scope, travel purpose, and travel mode are relatively limited [33]. To handle the above shortcomings, this paper proposes a method for obtaining travel mode label samples based on mobile phone travel trajectory data and residents' travel survey data.

The residents' travel survey data includes information like gender and the age of residents, the departure and arrival traffic zone number, and the departure time and arrival time of each travel record. The travel base station trajectory information extracted from MSD also contains the departure point and arrival point of each travel trajectory, the departure time and arrival time of each travel trajectory, and gender and age of the user corresponding to the IMSI number. Through connecting these attributes, the trajectory extracted from the MSD can be matched to the residents' travel survey data. Therefore, the travel mode information of the mobile phone travel trajectory can be obtained once the two records match successfully under the set rules, elaborated as follows.

To fuse these two types of data, the principal task is to establish a spatial correspondence between the base station and the traffic community. Then, the travel based on the mobile signaling data can be integrated into the travel based on the traffic community. Preliminary screening of mobile phone data can be conducted through the date of residents' travel survey data, and the departure and arrival of the community. Then, the travel time in each survey data is screened again. Finally, the data are matched according to gender and age attributes. For successfully matched records, the travel mode recorded in the travel survey data of residents can be regarded as the travel mode corresponding to the mobile phone travel path. Figure 2 shows the specific matching algorithm rules and procedures.

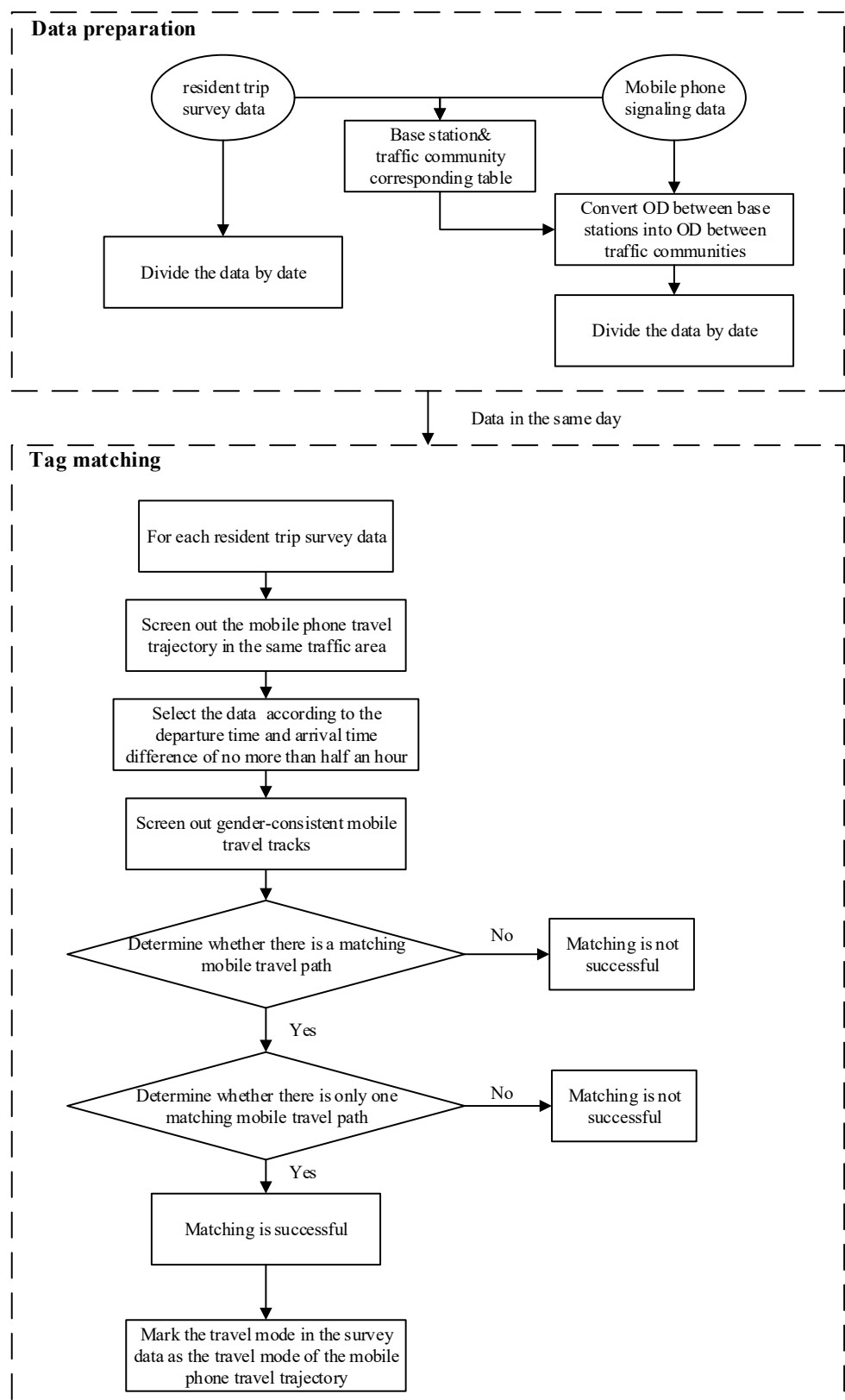

**Figure 2.** Flowchart for the sample tag matching.

*2.4. Feature Extraction and Selection*

The identification of the travel mode based on MSD is essentially a pattern recognition problem. The selection of features is a crucial step in the pattern classification problem, directly impacting on the

accuracy of the classification results. In the past, the extracted features in trajectory data mining were mostly limited to the temporal and spatial characteristics of the trajectory itself, and the data source was relatively simple. However, in the increasingly complex transportation system, the spatial and temporal distribution of different travel modes is getting smaller and smaller. It is often unsatisfactory to rely solely on the differences in the temporal and spatial characteristics to distinguish between different travel modes. Some scholars combined GIS in feature extraction and considered the differences among different travel modes in urban road network distribution, such as the similarity of travel routes and bus routes. However, few scholars were found to consider the relevant influencing factors involving traffic travel mode selection behavior and navigation path characteristics in their studies.

Different personal attributes, origin, and destination of the traffic in built environment affect the residents' travel mode choice behavior. The addition of characteristic parameters that affect residents' travel choice behavior may enhance the recognition capability of the model.

Moreover, in the case of different traffic conditions and departure times, the travel path and travel time required for each travel mode will be different. For example, in the morning and evening rush hours, the travel time required by cars on some congested roads is longer. To help consider the complexity of the traffic operating environment, Internet data was integrated into the feature extraction process. The travel distance and travel time between various travel modes were crawled from the Internet to help address the travel modes classification problem.

Following the analysis above, the study divided the features into three categories. The first category is the temporal and spatial features of the trajectory itself. It is also the characteristic of different travel modes in the travel trajectory. The second category is the personal attributes and environmental characteristics related to travel mode selection. Finally, the third category is navigation data which features considering paths and road conditions.

Seven temporal and spatial characteristics of the trajectory were extracted first. The travel distance "Distance" was calculated by distance of the travel track in the road network after map matching. The departure time "O_time" and the arrival time "D_time" were obtained from the track record in MSD, and travel time "Time" was obtained by subtracting these two characters. And the average speed "speed" was calculated by using "Distance" divided by "Time". The last step was defining the morning peak from 7 to 9 and the evening peak from 17 to 19 to judge whether the departure time and arrival time was at a peak or not in order to get the "O_peak" and "D_peak".

The personal attributes incorporated in the model include (i) the gender "Sex" by defining male as 1, female as 0; (ii) the age by defining 1 for travelers under 20 years old, 20~29 as 2, 30~39 as 3, 40~49 as 4, 50~59 as 5, and 60 years old or older as 6.

Then, we proceeded with the built environment features. A total of twenty features that can be divided into six categories were extracted. First is the coverage rate of a bus stop. After intersecting the 300 m and 500 m bus station buffer layer with the traffic zone layer in GIS, the area of the site coverage area corresponding to the service radius of each traffic community was obtained. Comparing the site coverage area of each traffic zone with the zone area, the 300 m bus station coverage rate and 500 m bus station coverage rate of the traffic community could be obtained. According to the spatial geographical location of the departure and arrival points of the track, four features including a 300 m coverage rate of a bus stop in the departure and arrival area "O_BS300" and "D_BS300", as well as a 500 m coverage rate of a bus stop in the departure and arrival area "O_BS500" and "D_BS500" corresponding to the trajectory can be obtained. Next were the bus line repetition coefficient, intersection density, road network density, and land mix in the departure and arrival area. Similar to the acquisition method of the coverage rate of a bus stop, their calculation methods are not described in detail here. What worth noting is the land use mixing index, which refers to the relative proximity of different functional land in the area. In the paper, the land use type was reclassified and divided into office, commercial, and residential according to the type of land use plan. The entropy index was used to measure the

mixing degree of land use, which can quantify the equilibrium of land use in a given area, and the calculation formula was shown as the following:

$$H_i = -\sum_{k=1}^{n} p(k) \ln(p(k))$$

(1)

where k represents the ratio of the area of the *k*th land type for the total area of the cell in the traffic zone *i* and *n* is the land type.

After feature extraction, feature selection, i.e., the selection of features from the original feature set to optimize the overall model performance, was carried out. Through the difference analysis of travel mode and travel behavior in the previous section, 29 features related to travel mode classification were extracted as the original features. After eliminating irrelevant and redundant features, the optimal feature subset was selected as an input to train the machine learning algorithms and models. In this study, filtered feature selection was adopted. The principle was to separately count the statistical indicators of each variable, according to which the characteristics were selected. This selection strategy is simple and easy to implement, and is beneficial to understanding the data. Chi-square test is often used for classification problems. The basic idea is to infer whether there is a significant difference between the overall distribution and the expected distribution based on the sample data or to infer whether the two categorical variables are related or independent.

$$x^2 = \sum \frac{(A - T)^2}{T}$$

(2)

where $A$ denotes the actual value, $T$ denotes the theoretical value.

The feature selection steps in the study are as follows in the Algorithm 2:

---

**Algorithm 2.** *Procedures for feature extraction.*

---

Step 1: Traversing the number of features *k*.
Step 2: With the univariate feature selection, the chi-square test is used to select the optimal *k* features.
Step 3: Use the combination of *k* optimal variables to construct the classification model and calculate the model accuracy.
Step 4: Choose the combination of variables under the optimal model accuracy as the selected feature.

---

### 2.5. Model Construction Based on Random Forest

Random Forest, also known as Random Decision Forest, is an ensemble learning algorithm composed of multiple decision trees and can be used for both classification and regression tasks. Firstly, through the bootstrap resampling technique, *k* samples are randomly selected from the original training sample set to form a new training sample set. Then, the obtained training samples are used to construct a decision tree, together forming a random forest. Finally, the classification result is determined by the classification tree. The random forest algorithm has the following advantages over other machine learning methods [34]: (1) It has high accuracy among the classification algorithms widely used at present. (2) The importance of each feature vector in the feature space can be evaluated by indicators like the Gini index. The importance measure for a particular variable is obtained as the average decrease of Gini impurity index over all trees in the forest. For a candidate splitting variable $X_i$ with a possible number of categories as $L_1, \ldots, L_j$, the Gini impurity index for this variable can be calculated. (3) It can effectively deal with large data sets and data samples with high dimensional characteristics, so it does not need to conduct dimensionality reduction. (4) The accuracy of the model

can be guaranteed even when there are many default data values. (5) The construction and execution speed of the model is fast and is not associated with the overfitting phenomenon.

$$G(X_i) = \sum_{j=1}^{J} P\big(X_i = L_j\big)\big(1 - P\big(X_i = L_j\big)\big) = 1 - \sum_{j}^{J} P\big(X_i = L_j\big)^2 \tag{3}$$

where $G(X_i)$ denotes the Gini impurity index for the variable $X_i$; $P\big(X_i = L_j\big)$ represents the estimated category $X_i = L_j$ probabilities. Once Gini impurity indices are calculated for each candidate splitting variable, the split is conducted on the variable that has the highest value.

The travel mode identification model based on the random forest algorithm mainly consists of four parts, namely the construction of training set and test set, the selection of feature vectors, the construction of decision tree, and voting to determine the optimal classification. The specific model construction process is as followed in Algorithm 3:

---

***Algorithm 3.*** *Procedures for model construction.*

---

**Step 1:** For the original travel path sample set, the samples were randomly divided following the ratio of 3:1 while 25% of the samples were taken as the test set.
**Step 2:** For the above-determined training sample set $D(x_1, x_2, ..., x_n)$, by taking $k$ as random repeatable samples, construct the random vector set $D_1, D_2, ..., D_k$.
**Step 3:** In the selected features, $m$ specific variables are randomly selected. Determine the optimal identification point using these variables.
**Step 4:** For each random vector $D_i$, construct a decision tree.
**Step 5:** Repeat step 3 and step 4. Finally, construct $k$ decision trees.
**Step 6:** For an input vector, each decision tree in the random forest votes.
**Step 7:** Finally, votes from all the decision trees are counted, among which the travel mode of this vector is the one with the majority of votes.

---

## 3. Data

### 3.1. Location Analysis

Our study area, Kunshan, is located inside the Yangtze River Delta economic area (YRDEA), one of the fastest-growing regions in China. From the picture, we could see that there are three of the biggest cities in China around Kunshan: Shanghai (to the east), Nanjing (to the west and the capital of Jiangsu province), and Hangzhou (to the south and the capital of Zhejiang province) [35]. It was selected for the studies in this paper, mainly because 4G telecommunication facilities are well established in its city area, which provides an ideal hardware/data condition for the topic addressed in this paper.

### 3.2. Data Description

#### 3.2.1. The Residents' Travel Survey Data

A traditional resident travel survey is the most basic way to obtain the residents' traffic demand for the local planning department. Despite the growing popularity and application of big data, the traditional residents' survey is still an indispensable way to obtain traffic information. A key advantage of residents' travel survey data is that they allow us to probe population mobility patterns together with rich sociodemographic information, which also makes them still one of the most reliable tools for transportation research [36].

In June 2017, we conducted a comprehensive traffic survey in Kunshan. The survey randomly selected 5150 households in urban areas and 250 households in peripheral villages, collecting 38,598 pieces of valid travel data, as shown in the Table 2.

**Table 2.** Residents' travel survey data examples.

| ID | Sex | Age | Order | Leave_ID | Arrive_ID | Leave_ID | Arrive_ID | Mode |
|----|-----|-----|-------|----------|-----------|----------|-----------|------|
| 1111 | Man | 30 | 1 | 504 | 510 | 7:20 | 8:01 | bus |
| 4289 | Woman | 29 | 2 | 329 | 504 | 6:29 | 7:31 | car |

Residents' travel survey data was used as a data set for known travel modes. Although the sample size was limited, sample labels can be extracted for unlabeled mobile phone data and used while modeling. Therefore, data matching between two sources is an important issue that needs to be solved in this research. On the one hand, the residents' travel survey data includes information like ID, gender, and age, as well as trip information like the starting traffic area ID, the arriving traffic area ID, leaving and arriving times, and travel modes of each record. On the other hand, the travel base station trajectory information extracted from the MSD includes the departure point and arrival point of each travel trajectory, departure time and arrival time, gender, and age of the users corresponding to the IMSI number. How to set rules to match these attributes and integrate MSD with the residents' travel survey data is a complex problem that needs to be solved in order to obtain the mobile phone trace sample data with travel modes.

3.2.2. The Mobile-Phone Signaling Data (MSD)

This study takes the MSD of Kunshan city as the primary data source. The daily data production volume is 65G, about 2.7 billion data, and the average person generates 1000 pieces of data per day, with an average update time of 20s/piece. When a mobile-phone user triggers a signaling event, the system will automatically record the user's signaling information, with one MSD sample containing multiple fields. To complete the matching work with the residents' travel survey data, MSD from June 12 was selected as the mobile phone data source for analysis. After pre-processing and identification of the parking point, the travel track information was extracted, obtaining a total of about 4.25 million data samples. The raw 4G MSD attributes used in the study are shown in Table 3.

**Table 3.** 4G MSD feature attributes.

| No. | Feature | Identification | Number | Feature | Identification |
|-----|---------|----------------|--------|---------|----------------|
| 1 | STAT_DATE | Date | 7 | BEGIN_TIME | Event start time |
| 2 | PROCEDURETYPE | Event | 8 | END_TIME | Event end time |
| 3 | MSISDN | Phone number(mask) | 9 | KEY_WORD1 | User status |
| 4 | IMSI | Phone unique identifier | 10 | CITY_CODE | Location city |
| 5 | LAC | Base location area coding | 11 | REGION | Phone number belonging city |
| 6 | CELL | Area coding | 12 | DAY_NUMBER | Tag data date |

The fields that are valuable for user trajectory information extraction and traffic analysis are listed as follows.

The unit of travel mode identification is a segment of travel trajectory. The acquisition of trajectory information from the mobile phone signaling is the primary task, as the original MSD is a continuous time-stamped base station location point. First, since there is a large amount of duplicated, invalid, and noisy data in the original MSD, it is necessary to filter and denoise the original data. Then, it is necessary to filter out the positioning information when the user is in the travel state to obtain the approximate travel trajectory of the user. As this topic has been thoroughly discussed in past research, this article will not go into depth in discussing it. The default MSD is the station-based user travel trajectory.

However, to improve the accuracy of the mode recognition, the travel trajectory connected by such a base station is insufficient. Therefore, this paper aims to match the travel trajectory based on the base station to the actual road network to form a real travel trajectory based on the road intersection. It enables extracting high-quality features in order to improve mode identification. How to achieve map matching based on the travel trajectory of the base station is another challenging task that is solved in this paper.

---

*Valuable MSD fields*

---

**STAT_DATE:** The date on which the MSD is generated.

**IMSI:** International Mobile Subscriber Identification Number. It is a unique international user identification code assigned by the system to each mobile user. Further, it is also the unique identifier of the signaling data subject. From this field, relevant registration information such as the gender and age of the user can be obtained. One of the main advantages of MSD in the field of traffic analysis is anonymous data processing, meaning there are no user privacy and information security issues.

**LAC/CELL:** A physical base can be uniquely found by the LAC and CELL to match its latitude and longitude information.

**BEGIN_TIME/END_TIME:** The start time and the end time of the signaling event. In general, the occurrence of signaling events and information propagation is rapid. The two fields can be merged into a single field, indicating when a signaling event occurred in the data pre-processing stage.

**PROCEDURE TYPE:** Different event type numbers triggered by signaling.

---

### 3.2.3. AMAP Data

The classification used to rely on static data like residents' travel survey data. In recent times, GPS data and MSD are more frequently used. However, these data are identified from the perspective of the trajectory itself. This paper innovatively combines travel trajectory data with navigation data. Using the official application programming interface (API) interface, the travel distance and travel time of various travel mode can be connected. The data obtained is the simulation data generated in the real road network environment, considering the real-time traffic condition, which is significant to travel mode identification.

The path planning interface in the AMAP API was used in this study. Travel distance and travel time required for the navigation path using walking, bicycle, bus, and car could be crawled separately once the OD point of the trip is imported into AMAP. In general, there are several recommended paths to select for each travel mode. This article selected the best-recommended path for navigation when crawling. Further, considering the volatility of road conditions in a single day, two sets of data in flat peak and peak periods were crawled. Then, the corresponding navigation path feature parameters were determined according to the occurrence time of the travel trajectory. Table 4 is an example of crawled data in AMAP. O_base and D_base represent the departure and arrival base station ID through which the latitude and longitude information could be extracted. It is then possible to obtain the travel distance and travel time of the navigation recommended route for different travel modes.

**Table 4.** Recommended path navigation data crawling example.

| O_base | D_base | Walk-T | Walk_D | Bike-T | Bike-D | Bus-T | Bus-D | Car-T | Car-D |
|--------|--------|--------|--------|--------|--------|-------|-------|-------|-------|
| 415 | 2985 | 13821 | 17276 | 4455 | 18561 | 4680 | 18664 | 2300 | 27128 |
| 2985 | 415 | 13830 | 17287 | 4425 | 18438 | 4244 | 18260 | 2256 | 25535 |
| 1427 | 404 | 3070 | 3837 | 923 | 3846 | 2329 | 6702 | 681 | 4174 |
| 404 | 462 | 2868 | 3585 | 860 | 3582 | 1873 | 4218 | 747 | 4142 |
| 462 | 1427 | 1280 | 1600 | 496 | 2065 | 1622 | 2668 | 599 | 2080 |
| 508 | 2751 | 2899 | 3624 | 954 | 3977 | 2746 | 7320 | 593 | 4025 |
| 2751 | 2048 | 3179 | 3974 | 956 | 3982 | 2683 | 4781 | 618 | 4832 |
| 2048 | 1584 | 7006 | 8758 | 2162 | 9010 | 3289 | 8789 | 1223 | 9268 |
| 1584 | 2985 | 6648 | 8310 | 2030 | 8459 | 3182 | 14559 | 1231 | 8999 |

## 4. Implementation and Results

*4.1. Model Effect Evaluation*

To evaluate the performance of the model in mode identification, the precision rate, recall rate, and F-score are selected as evaluation indexes in this study [37]. For a classification problem, the sample can be divided into true positive (TP), false positive (FP), true negative (TN), and false negative (FN) according to the combination of its real category and the prediction category. The precision rate refers to the proportion of the correct number of travel mode samples identified by the model to the total samples identified as the travel mode.

$$P_i = \frac{TP_i}{TP_i + FP_i} \tag{4}$$

where $P_i$ denotes the precision rate of mode $i$, $TP_i$ represents the correct number of samples of mode $i$ in the model, and $FP_i$ represents the number of incorrect samples of mode $i$.

Recall rate refers to the proportion of the correct sample number of a particular travel mode identified by the model to the actual travel mode.

$$R_i = \frac{TP_i}{TP_i + FN_i} \tag{5}$$

where $R_i$ denotes the precision rate of mode $i$, and $FN_i$ stands for the number of incorrect samples of mode $i$.

F-score is a commonly used evaluation index in classification problems. The precision rate and recall rate are a set of contrasting indicators. When the precision rate is high, the recall rate is low. In actual model evaluation, precision or recall alone is not enough. Therefore, another statistical index composed of the ratio of recall and precision is defined and formulated as follows:

$$F - score = \frac{2 \times P_i \times R_i}{P_i + R_i} \times 100\% \tag{6}$$

The definitions of the parameters in Equations (2)–(4) are shown in Table 5 below.

**Table 5.** Definition of parameters in the model evaluation indexes.

| Actual Sample \ Identify Sample | Number of Samples Identified as $i$ | Number of Samples Identified as Other Classes |
|---|---|---|
| Number of class $i$ samples | $TP_i$ | $FN_i$ |
| Number of other samples | $FP_i$ | $TN_i$ |

*4.2. Sample Data Matching*

The travel modes in the residents' survey data can be divided into: walking, electric vehicle, taxi, motorcycle, bus, bicycle, unit bus, private car and unit matching car, unit business car, etc. The usage proportion for each travel mode is shown in Figure 3 below.

It can be seen from Figure 4 that private cars and electric cars have a high proportion of usage, i.e., 29.07% and 28.37% respectively, followed by walking, bus, and bicycle. According to the travel structure of Kunshan city and the driving characteristics of vehicles, the travel modes were divided into walking, bicycle, electric vehicle, bus, and car. Note that, here, bicycles include public bicycles and other bicycles, and cars include taxis, private cars and online car-hailing. Due to the low proportion of other travel modes, the differences among the above-mentioned travel modes are not apparent, thus are not differentiated. By combining and excluding the travel records of other travel modes, a total of 36,987 valid survey data samples were obtained.

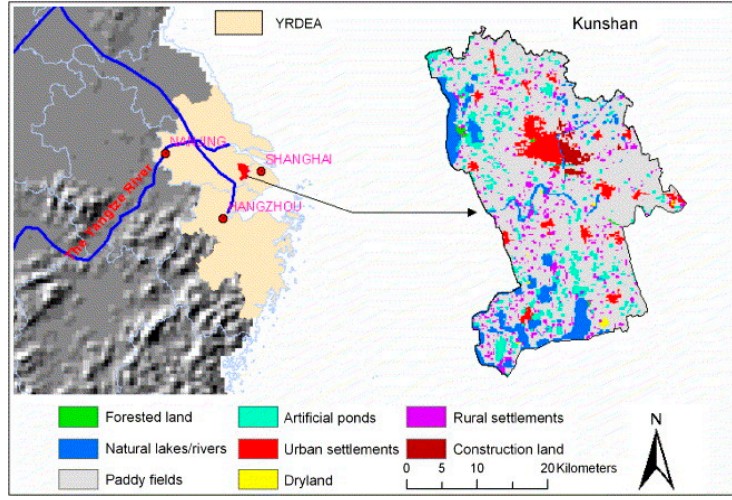

**Figure 3.** Kunshan's geographical location.

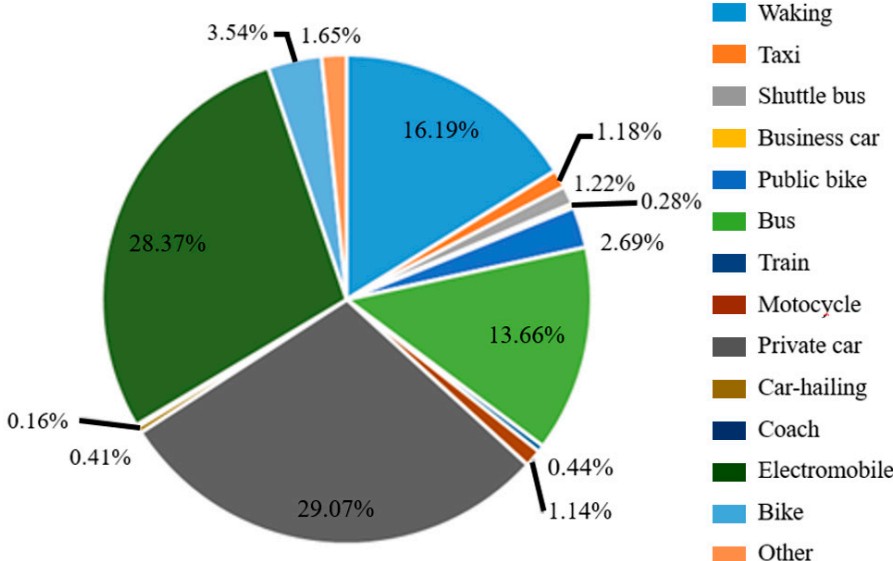

**Figure 4.** Structural ratio of different travel modes.

The distribution of these sample data in Kunshan is presented in Figure 5. After matching, a total of 5861 mobile phone travel track data were obtained, including 1147 walking samples, 570 bicycles samples, 1562 electric vehicles samples, 732 buses samples, and 1850 cars samples. These numbers were consistent with the mode share obtained from the residents' survey data. Partially matched data are shown in Table 6. Id represents the serial number of residents' travel survey data, and type represents the travel mode type. O_time and D_time represent the departure time and arrival time filled in the survey data. O_id and D_id represent the number of traffic area at the starting and ending points. Leave_time and arrive_time represent the departure time of the first stop point (starting point) and the arrival time of the second stop point (ending point) in the successfully matched trajectories.

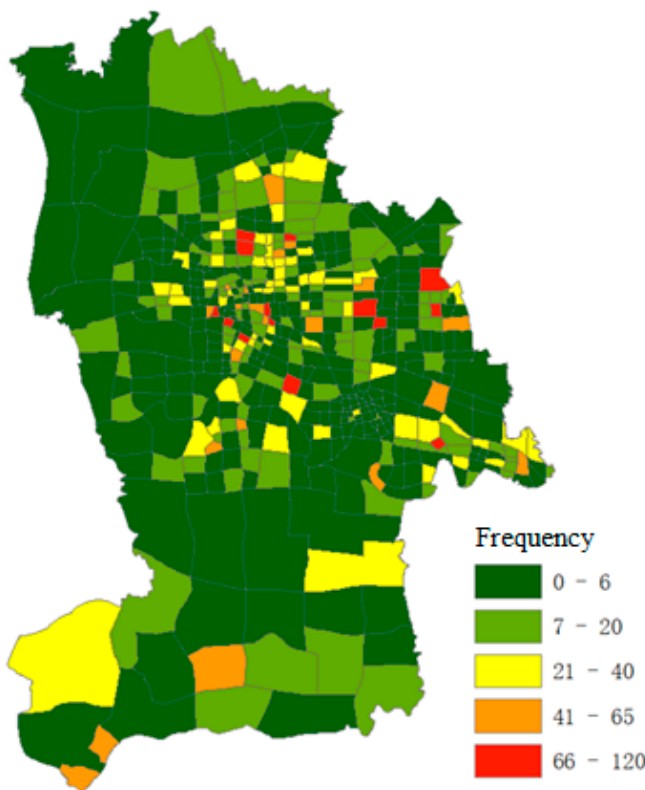

**Figure 5.** Spatial distribution of the selected sample.

**Table 6.** Matching data scheme.

| ID | Type | O_Time | O_id | D_Time | D_id | Sex | Age | Leave_Time | Arrive_Time |
|---|---|---|---|---|---|---|---|---|---|
| 20002 | 3 | 18:15:00 | 420 | 18:50:00 | 164 | 1 | 38 | 18:17:54 | 18:56:53 |
| 9002 | 3 | 7:40:00 | 384 | 8:03:00 | 396 | 2 | 26 | 7:38:16 | 8:02:22 |
| 26002 | 5 | 11:24:00 | 285 | 11:48:00 | 270 | 2 | 35 | 11:26:51 | 11:42:53 |
| 15002 | 3 | 15:30:00 | 7 | 15:40:00 | 7 | 2 | 58 | 15:29:26 | 15:39:27 |
| 2002 | 1 | 9:22:00 | 323 | 9:40:00 | 324 | 2 | 60 | 9:24:49 | 9:34:35 |
| 12002 | 3 | 17:00:00 | 149 | 17:30:00 | 60 | 2 | 51 | 16:51:12 | 17:20:59 |
| 16002 | 1 | 8:40:00 | 27 | 9:00:00 | 27 | 2 | 58 | 8:40:17 | 8:59:54 |
| 31002 | 5 | 17:00:00 | 64 | 17:30:00 | 430 | 1 | 47 | 16:54:58 | 17:25:56 |
| 33002 | 3 | 17:00:00 | 206 | 17:29:00 | 81 | 2 | 29 | 16:53:16 | 17:21:24 |
| 36002 | 1 | 21:42:00 | 243 | 21:49:00 | 243 | 2 | 59 | 21:44:49 | 21:50:25 |

### 4.3. Feature Extraction and Selection

According to the feature selection process above, the relationship between the accuracy of the model and the number of features is plotted. As shown in the Figure 6, the accuracy of the model increases with the number of features. When the number of features is greater than 22, the model accuracy remains constant, and it reaches the highest level when the number of features is 24. The corresponding features are: speed, O_BS500, D_BS300, O_BS300, bus_distance, car_distance, car_time, bus_time, O_RJdensity, D_RJdensity, time, bike_distance, distance, walk_distance, age, D_BLrepetition, bike_time, D_landmixing, walk_time, O_landmixing, O_BLrepetition, O_RDdensity, D_RDdensity, and O_peak.

Figure 7 presents the importance ranking of 24 features in the model based on the random forest algorithm. It can be seen from the figure that speed is the most significant feature in the model. The importance ranking of features is travel trajectory spatio-temporal features, followed by navigation path features, and travel mode selection features. Moreover, the distance attribute is found to be more significant than the time attribute in navigation features.

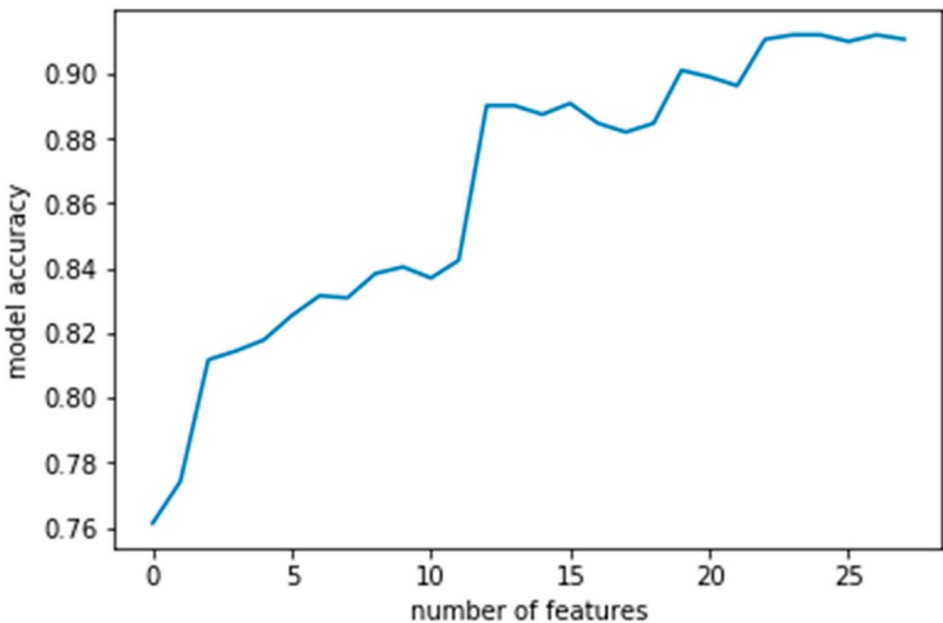

**Figure 6.** Comparison of model accuracy with the number of features.

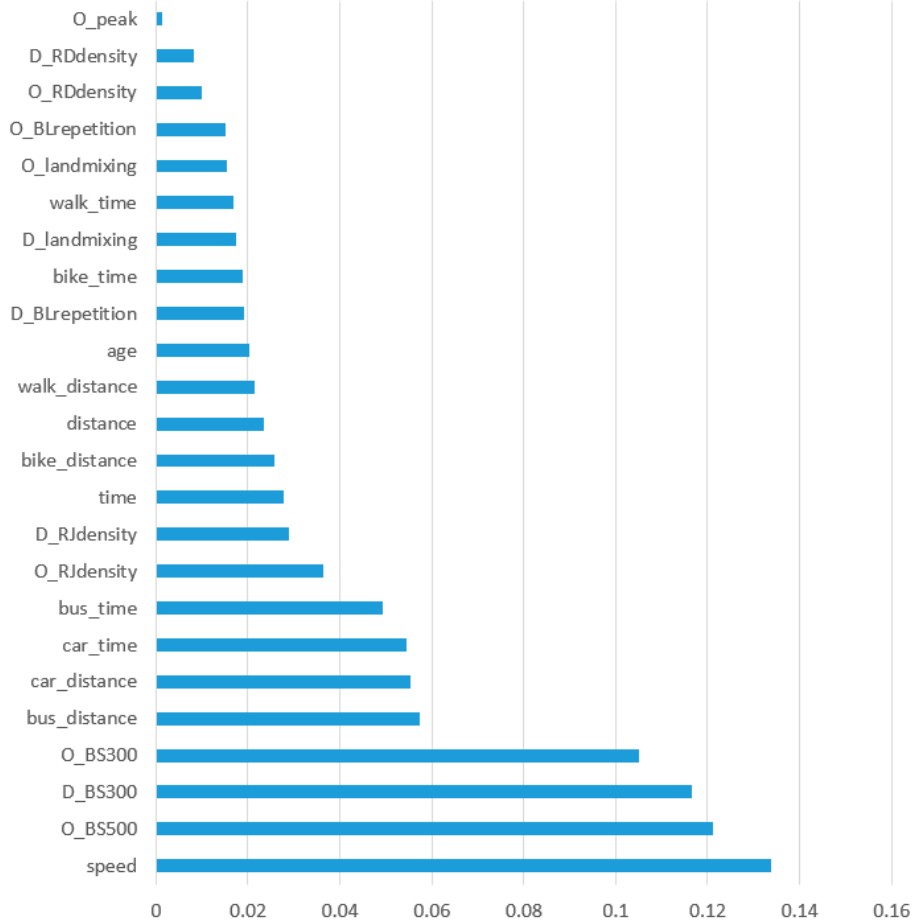

**Figure 7.** Importance ranking of feature parameters.

## 4.4. Model Accuracy Analysis

The model is coded in Python using the third-party machine learning module, scikit-learn (sklearn). Sklearn module encapsulates commonly used machine learning algorithms, including regression,

dimensionality reduction, classification, clustering. It is a simple and efficient data mining and data analysis tool [38]. To ensure the reliability and generalization performance of the model results, this paper randomly divides the samples in the ratio of 3:1. Three-fourth of the samples were used as model training samples while the remainder were used as test samples.

Table 7 is the test set confusion matrix and evaluation index based on the random forest algorithm.

**Table 7.** Random forest algorithm identification results.

| Identified Travel Mode | Confusion Matrix | | | | | | Evaluation Index | | |
|---|---|---|---|---|---|---|---|---|---|
| | Real Travel Mode | | | | | | | | |
| | Walk | Bike | Electric-Bike | Bus | Car | Total | Precision Ratio | Recall Ratio | F1 |
| Walk | 256 | 6 | 0 | 0 | 0 | 262 | 0.98 | 0.91 | 0.94 |
| Bike | 5 | 121 | 1 | 16 | 0 | 143 | 0.85 | 0.84 | 0.84 |
| Electric-bike | 10 | 10 | 394 | 60 | 0 | 474 | 0.83 | 0.97 | 0.89 |
| Bus | 10 | 6 | 11 | 93 | 0 | 120 | 0.78 | 0.53 | 0.63 |
| Car | 0 | 1 | 3 | 5 | 458 | 467 | 0.98 | 1.00 | 0.99 |
| Total | 281 | 144 | 409 | 174 | 458 | 1466 | 0.90 | 0.90 | 0.90 |

Concerning the model accuracy, we made a comparison between the results of previous research and ours. Yamada et al. (2016) used a dataset from Scenargie software [31], and the recall of their model for car mode was 0.75. Xu et al. (2011) evaluated their method with 500 mode sequences labeled by individual users, with 80% of the data for training and the other 20% for testing [39]. Their F1 values was 0.89. Li et al. (2017) used a labeled dataset consisting of 7 days data of 10 users to evaluate their method [40]. The reported average precision, recall and F1 values were around 0.64–0.91, 0.74–0.81, and 0.72–0.83, respectively, depending on the parameters. It is clear that the proposed model outperform those in existing research. Among the 1466 test set samples, 1322 samples were correctly identified through the random forests model, achieving an overall accuracy of 90.2%. The accuracy of walking and car is up to 98%. Walking travel is limited by people's physical strength, and its travel speed and distance are obviously different from that of other travel modes. Therefore, the recognition accuracy of walking is relatively ideal. The recognition accuracy of the car is also very high due to its navigation path feature aided discrimination. Further, its speed is also different from other travel modes. However, the precision rate and recall rate of bus trips were only 78% and 53% respectively, which is far lower than the average accuracy of the model. The main reason for the poor performance in bus identification may be the existence of too many uncertainty, for example, random bus users, uncertain waiting and stopping time, and the inability to conduct path adjustment according to traffic conditions. The precision rates of the bike and electric bike were both around 85%. These two modes are likely to be identified as a bus because with the increase in the share of non-motor vehicles, since their travel speed and travel distance are close to those of a bus.

## 5. Conclusions

In this research work, we have established a travel mode identification model based on mobile signaling data, combined with residents' travel survey data, GIS data, and navigation data, aiming to accurately identify the travel modes as well as travel trajectories of urban travelers/commuters. The social values of works in this paper are twofold. First, imbalance between transport supply and demand is the core factor in congestion as well as many other transport externalities. Thus, for the sustainable development of an urban transport system (and also the vibrancy of our livable cities), to manage travel demand and wisely plan/construct the supply are highly significant. In order to make optimal decisions/policies on transport demand and supply, it is an important prerequisite to accurately obtain the multimodal travel demand (that is the travel mode of each traveler). Considering that existing methods on the travel mode detection are not accurate, this work based on big data is an important new pathway to solving this problem. Second, to accurately detect the travel modes of urban travelers are also important to resolve many existing social problem; for instance, the special

aids and cares for elderly travelers. If we can detect their most commonly used travel modes and trip chain, the transport authorities can improve the placement and arrangements of the facilities.

This study built a travel mode identification model based on the 4G MSD, the residents' travel survey data and the GIS data in Kunshan city, as well as the navigation data crawled from AMAP API. The travel mode labels of mobile phone track samples were obtained using the attribute matching rule with the mobile phone track. Regarding model feature extraction, 29 features were extracted from the spatial and temporal characteristics of trajectories, travel mode selection behavior characteristics, and navigation path characteristics. The precision rate of the model reached 90.2%, and the recognition result of walking and car is satisfactory. However, the recognition accuracy of bus is low and needs further optimization.

Two aspects need to be further investigated to overcome these issues. First is the identification of the trajectory switching point and division of the travel segments. This paper takes the trip as the unit for analysis, taking into account the dominant mode of travel only by priority. In future research, the concept of a transition point can be further incorporated, i.e., by dividing the travel segment by the changes in the travel mode, which provides proper path matching and in-depth mining of trajectory data. The path matching algorithm proposed in this paper does not match the roads lacking intersections and those with vertical parallel paths such as overpasses and tunnels. Further, the path matching algorithm can be improved by integrating elevated entrances and exits. Moreover, in the process of feature extraction, the trajectory information is worth further exploration, including for similarities with the bus network and the similarities between the navigation paths of different travel modes.

**Author Contributions:** Formal analysis, C.A.; Investigation, Z.L. (Zhen Long); Methodology, Z.L. (Zhenbo Lu); Resources, Z.L. (Zhenbo Lu); Software, C.A.; Supervision, J.X.; Validation, J.X.; Visualization, Z.L. (Zhen Long); Writing—original draft, Z.L.; Writing—review & editing, Z.L. (Zhenbo Lu).

**Funding:** This research was funded by the National Natural Science Foundation of China (71971060).

**Acknowledgments:** The authors gratefully acknowledge assistance with Residents' travel survey data and mobile phone signaling data from Kunshan Land Resources Bureau.

**Conflicts of Interest:** The authors declare no conflict of interest.

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
