# Peer review of "A Random Forest Model for Travel Mode Identification Based on Mobile Phone Signaling Data"

_sustainability, doi:10.3390/su11215950_

Round 1
Reviewer 1 Report
Summary:
The paper "A Random Forest Model for Travel Mode 2 Identification Based on Mobile Phone Signaling Data" aims at the identification of the
travel mode of users, mainly based on smartphone data. In addition, data from surveys are considered for the travel mode identification.
The identification, in turn, is based on machine learning. More precisely, a random forest approach is applied to the aforementioned data
setting. The authors of the paper describe related work, relevant background information, and how the analyzed data was obtained. Then,
the machine learning approach is described, followed by the results of the machine learning application. The authors conclude that for
particular travel mode categories (e.g., bike), the results reach a very high accuracy level. They also conclude that mobile signal data
was less used by other authors in this context, what is also the reason for their results.
Points in favor:
- The paper fits to the scope of the journal
- The paper deals with a topical subject
- The paper uses real-life data
- The paper shows experimental results
- The paper discusses related works
- In general, the paper shows a common thread
Points against the paper:
- The paper is very difficult to read, what is based on several reasons. Some examples are provided, e.g.,
(1) Many statements are used without any introduction of them or relevant information before, e.g.,
(1,1) Abstract: GIS is only used as an abbreviation, even if such an abbreviation is more
or less clear, it must be introduced
(1,2) Line 59: The traditional four-step approach -> never introduced before
(1,3) Line 96: AMAP never introduced, also the API later on
(1,4) Lines 13 and 14: What is medium and macro level in this context?
...
(2) Language issues can be found, e.g.,
(2,1) Abstract: Line 18: expect buses -> except buses
(3) Usage of abbreviations
(4) Formatting of Table 1
(5) Blanks are missing in captions, e.g., 2.1.1
(6) Many statements that include key messages are not supported by references, e.g.,
Line 30: "... are not detectable ..." -> Reference or explanation needed
...
- Many sentences must be scientifically improved, e.g.,
(1) Line 130: "... relevant scholar at home ..."
(2) Line 165: "... are relatively complement ..."
(3) Line 166-168: Why should this cause randomness?
...
- Limitations of the study must be discussed in a separate section
- Why this machine learning approach and not others?
- Less data to the surveys are provided -> descriptive statistics?
Author Response
Comment 1-1: The paper "A Random Forest Model for Travel Mode 2 Identification Based on Mobile Phone Signaling Data" aims at the identification of the travel mode of users, mainly based on smartphone data. In addition, data from surveys are considered for the travel mode identification. The identification, in turn, is based on machine learning. More precisely, a random forest approach is applied to the aforementioned data setting. The authors of the paper describe related work, relevant background information, and how the analyzed data was obtained. Then, the machine learning approach is described, followed by the results of the machine learning application. The authors conclude that for particular travel mode categories (e.g., bike), the results reach a very high accuracy level. They also conclude that mobile signal data was less used by other authors in this context, what is also the reason for their results.
Points in favor:
- The paper fits to the scope of the journal
- The paper deals with a topical subject
- The paper uses real-life data
- The paper shows experimental results
- The paper discusses related works
- In general, the paper shows a common thread
Response: Thanks for your encouragement and recognition of our work. Your comments and suggestions have been carefully investigated and are quite helpful for the amendments of this manuscript.
Comment 1-2: The paper is difficult to read, what is based on several reasons. Some examples are provided, e.g.,
(1) Abstract: GIS is only used as an abbreviation, even if such an abbreviation is more or less clear, it must be introduced GIS is only used as an abbreviation, even if such an abbreviation is more or less clear, it must be introduced
(2) Line 59: The traditional four-step approach -> never introduced before
(3) Line 96: AMAP never introduced, also the API later onAMAP never introduced, also the API later on。
(4) Lines 13 and 14: What is medium and macro level in this context?
Response: Thank you for your comment. We apologize for these confusing parts in the previous verion of the paper. In accordance with your comments, we made the following modifications.verion of the paper. In accordance with your comments, we made the following modifications.
All the abbreviations including GIS, AMAP, API have been explained before using. The traditional four-step approach refers to the: Trip Generation, Trip Distribuiton, Mode Choice, and Route Choice, which are used in the existing software packages/tools for transport system analysis and flow predictions (Sheffi, 1985). was not necessarily introduced here. All we want to express was that the traditional way to identify the transportation mode was using residents’ survey data. So we have revised this sentence as follow: “The traditional method uses residents’ survey data.” AMAP is a web map, navigation and location based services provider in China where we could extract navigation path characteristics to identify the transportation mode and we have explained in line 104. The API is a tool to help us crawl the required information from AMAP and we have explained in line 222. it is really true as the Reviewer suggested that the medium and macro level are difficult to define. We want to express that MSD is mostly used at network analysis, mobility analysis, event detection, and urban planning(Blondel et al., 2015; Calabrese et al., 2015; Naboulsi et al.,2016; Wang et al., 2018; Yuan and Raubal, 2016)but our previous expression seemed not suitable. So we have revised this sentence as follow: “However, due to their noisy and temporally irregular nature, extracting mobility information such as transport modes from these data is particularly challenging.”
References:
Blondel, V.D., Decuyper, A., Krings, G., 2015. A survey of results on mobile phone datasets analysis. EPJ Data Sci. 4, 10.
Calabrese, F., Ferrari, L., Blondel, V., 2015. Urban sensing using mobile phone network data: a survey of research. ACM Comput. Surv. 47 (2), 1–20.
Naboulsi, D., Fiore, M., Ribot, S., Stanica, R., 2016. Large-scale mobile trafc analysis: a survey. IEEE Commun. Surveys Tutorials 18, 124–161.
Wang, F., Chen, C., 2018. On data processing required to derive mobility patterns from passively-generated mobile phone data. Transport. Res. Part C: Emerg. Technol. 87, 58–74.
Sheffi, Y. 1985. Urban Transportation Networks: Equilibrium analysis with Mathematical Programming Models. Prentice-Hall, INC, Englewood Cliffs, New Jersey.
Yuan, Y., Raubal, M., 2016. Exploring georeferenced mobile phone datasets – a survey and reference framework. Geography Compass 10, 239–252.
Comment 1-3: Language issues can be found, e.g., Abstract: Line 18: expect buses -> except buses
Response: Thanks for the comment. We sincerely apologize for these typos and mistakes. To secure the high quality of this paper, we have incited a professional English editor to polish and proofread the manuscript, and also cearefully double checked the paper for any existing typos. We believe tha the quality of this paper is inherently improved.
Comment 1-4: Usage of abbreviations
Response: Thanks for your comment. This issue of abbreviations has been carefully solved in the new version of the paper, where the full names are provided, and suitable abbreviations are used only in the necessary occasions.
Comment 1-5: Formatting of Table 1
Response: Thanks for the comment. The formatting problem of Table 1 has been solved.
Comment 1-6: Blanks are missing in captions, e.g., 2.1.1
Response: Thanks for the comment. The formatting of captions has been corrected.
Comment 1-7: Many sentences must be scientifically improved, e.g.,
(1) Line 130: "... relevant scholar at home ..."
(2) Line 165: "... are relatively complement ..."
(3) Line 166-168: Why should this cause randomness?
Response: We sincerely thank you for the insightful comments. We have carefully checked these issues and inappropriate phrases previously used, and corrected them in the updated versions of the manuscript for your perusal.
Reviewer 2 Report
The paper is interesting, but presentation must be highly improved. For example, why section "Literature review" is placed in the middle of the Introduction, but without any number or label?
Besides, section 3 "Methodology" is not a methodology description, but a description of a technical proposal. Please, review sections and titles and reorganize the content in a more coherent way.
Figures and Tables are interesting, but experimental Section should be extended. In particular, Figure 5 should be extended to include state of the art technologies. Does the proposed model improve the precision of the existing models? For me that's a key question that must be answered.
Author Response
Comment 2-1: The paper is interesting, but presentation must be highly improved. For example, why section "Literature review" is placed in the middle of the Introduction, but without any number or label?
Response: Thank you for your encouragement and recognition of this paper. Your comments and suggestions have been carefully investigated and are quite helpful for the amendments of this manuscript. We have carefully addressed each comment one by one and the corresponding amendments are updated in the revised manuscript. To guarantee the high quality of this paper, we have incited a professional English editor to polish and proofread the manuscript, and also cearefully double checked the paper for any existing typos. We believe tha the quality of this paper is inherently improved.
Comment 2-2: Besides, section 3 "Methodology" is not a methodology description, but a description of a technical proposal. Please, review sections and titles and reorganize the content in a more coherent way.
Response: Thank you for the comments. The structure of the paper indeed has some problems. The core technologies and innovations focus on the use of map matching to obtain more useful features and data fusion to obtain labels. So first, the structure of the paper has been optimized. Problem description section and methodology section have been merged. In the new data section , the location has been analysed and the data have been decribed. Second, the methodology selection has been improved. More formulas and connotations have been added to support the theoretical basis of the paper.
Comment 2-3: Figures and Tables are interesting, but experimental Section should be extended. In particular, Figure 5 should be extended to include state of the art technologies. Does the proposed model improve the precision of the existing models? For me that's a key question that must be answered.
Response: Thanks for the acknowledgements by the reviewer.
There are indeed deficiencies in the feature selection. Our research could be extended to two aspects.The impact of the 24 features on the identification of transportation modes and the relationship between them are lack of research. What’s more, it can be seen from Figure 5 that after increasing the number of features to 21, the accuracy of the model remains basically the same. So whether the accuracy can be slightly reduced to improve the implementation efficiency is the direction of further research in the future.
The research on whether to improve the existing recognition accuracy was indeed lacking in this paper. So some other research have been mentioned in section 4.3. The accuracy of the identification of transportation mode especially for the car mode has been improved.
Reviewer 3 Report
First of all, English should be enhanced with the "Certificate of Professional English Proof reading Services". I think MDPI may provide this service. For example, title is not readable. "Random forest", what it means? It is terrible! Title should be condensed but with clear meaning. It failed in this paper. More over, in abstract, "---The accuracy levels were satisfactory for most of the modes expect buses." -> I cannot understand this part as well. There are numerous English grammar and typing errors. In general, the paper does not seem as an article, but just a report from the data summary. The logical structure is very weak, and there is no unique contribution in methodologies, and/or unique implications and suggestion. Even the paper does not fit into this journal. For example, the citations should be numbered with the matching number in references. - Introduction is too short and too confused with many ideas. "Literature review" should be separated with introduction. Introduction should show much more research background with "questions" and the logical structure of the paper with its objectives. It is too short to understand the "Unique contribution" of the paper. For example, it should be clear why the authors choose "Kunshan" city. There was no explanations on this decision on Kunshan. Frankly, as a foreigner, I do not know where Kunshan is located, and why we need to analyze Kunshan as "The Representative region of China". Literature review is too short as well to get the ideal model for the research. The authors did not show the appropriate reason of MSD model. It should be based on the comparative analysis on the previous papers with the strength and weakness of MSD compared with other methodologies. The authors chose "4G MSD attributes" arbitrary and subjective reason. It should be logically supported by the other papers as well as full explanations of the selection. There are too much jargon type of terminology such as "What worth noting is the land mixing." -> Explain the definition and conceptual characteristics of land mixing in detail. "To evaluate the accuracy of the model and its performance in mode identification, precision rate, 413 recall rate, and f-score is selected as evaluation indexes in this study." -> It is arbitrary and subjective criteria. We need citation s from other papers as well as the clear explanations. The most important criteria in the methodology is the statistical significance and/or mathematical as well as logical reliability. How can the authors say "The overall accuracy of the model reached 90%"?Author Response
Please see the attachment.

Reviewer 4 Report
Needs citation at points like line 43, the early 1950s (cite).
Line 45: namely statistical-based aggregation methods (cite) and non-aggregation methods based on probability theory (cite)
Line 56 - cite at the end of the sentence "A critical factor....".
Figure #4 is hard to read. Please make the color boxes in the legend bigger and easier to recognize.
Lione 465 "online taxi" do you mean a taxi called using the Internet? The phrase online taxi makes no sense, please correct.
Overall, the paper needs another pass in English editing. Nothing major but minor errors are distracting.
Author Response
Comment 4-1: citation problem:
Line 43:needs citation at points , the early 1950s.
Line 45: namely statistical-based aggregation methods (cite) and non-aggregation methods based on probability theory
Line 56: cite at the end of the sentence "A critical factor...."
Response: Thanks for your encouragement and recognition of our work. Your comments and suggestions have been carefully investigated and are quite helpful for the amendments of this manuscript. We apologize for these unscientific parts in the previous verion of the paper. The references have all been added. We believe tha the quality of this paper is inherently improved.
Comment 1-2: Figure #4 is hard to read. Please make the color boxes in the legend bigger and easier to recognize.
Response: Thanks for your comment. Figure #4 has been redrawn to make it more readable.
Comment 1-3: Lione 465 "online taxi" do you mean a taxi called using the Internet? The phrase online taxi makes no sense, please correct.
Response: Thanks for the comment. Just the same as you understand, the phrase has been changed into “online car-hailing”
Round 2
Reviewer 1 Report
Dear authors,
thanks for the hard work. However, still severe issues can be found and several of my issues have not been addressed.
Readability is still an issue, only some examples:
Line 8: individual traveller -> individual travellers
Line 27: to accurate predict -> accurately predict
Line 27: in light -> in the light
Line 93: Though systematic ... -> not a correct sentence
Line 110: Section 2 presents ... -> not a correct sentence
Table 1: badly formatted and difficult to read
Algorithm 1: difficult to read -> I would rather recommend a figure
-> Pseudocode would better for all algorithms
Limitations are not separately considered, also the justifications for the used machine learning approach are not considered. For a paper that uses ML as a basic pillar, these considerations must be accomplished.
Reviewer 3 Report
I appreciate the authors' efforts to enhance the quality of the paper, but it does not made full reflection of the comments. Rather, the authors made a terrible excuse of their logic. It is not appropriate attitude. As already pointed out, the abstract should show the real contribution of the paper in its implications and/or methodologies. But it does not show any of these. Using "Mobile-phone Signaling Data (MSD)", they found "the model achieved a high accuracy of 90% in Kunhsan city". The authors just explains the MSD is the best to find out the locatio of traveller's. So what? The social research paper should "examine" the social problems or difficulties. There is no such trobles to solve in this paper. The authors just explains the statistical reliability of MSD and its unique characters. It is just "report", because it does not give any solution for Chinese government, MSD users, or developers. They concludes that "Two aspects need to be further investigated to overcome these issues. First is the identification 546 of the trajectory switching point and division of the travel segments.---Further (actually it should be "second"), the path matching algorithm 552 can be improved by integrating elevated entrances and exits." -> this conclusion should come from the manual of MSD developing companies, not from social science researchers. There is no social/economical implications and suggestions in the paper. They just explained the technological characters of MSD. "This study aims to establish a travel mode identification model based on mobile signaling data" -> It is just a technical report for the new technology of MSD. There is NO "sustainable" issue at all. And thus the paper does not fit into our journal, Sustainability. The authors should submit the paper in IT related journals.Author Response
Please see the attachment.

Round 3
Reviewer 1 Report
The paper has been improved regarding the language. I still struggle with some issues mentioned before. In particular, mentioned in the last review, which was not tackled, namely the justification of the choice of the ML-method. In addition, I still miss the limitation section.
Then, still language issues and inaccuracies can be found:
New text in Section 5 says "... we have endeavored to evaluate and contrast all the existing types of big data ...", which is very ambitious, I guess the authors would state in the context of ....
The paper still needs major revisions.
Author Response
Comment 1-1: The paper has been improved regarding the language. I still struggle with some issues mentioned before. In particular, mentioned in the last review, which was not tackled, namely the justification of the choice of the ML-method. In addition, I still miss the limitation section.
Response: Thanks for your encouragement and recognition of our work. Your comments and suggestions have been carefully investigated and are quite helpful for the amendments of this manuscript. We apologize for this question. This is indeed a very important question. Actually, we explained this question in the response report but didnot add it to the text. We’ve added this part to the beginning of the literature review section (line 44-60).
Comment 1-2: Then, still language issues and inaccuracies can be found:
Response: Thanks for your comments. To secure the high quality of this paper, we have invited a professional English editor to polish and proofread the manuscript, and also cearefully double checked the paper for any existing typos. We believe that the quality of this paper is inherently improved.
Comment 1-3: New text in Section 5 says "... we have endeavored to evaluate and contrast all the existing types of big data ...", which is very ambitious, I guess the authors would state in the context of ....
Response: Thanks for your comments. The wording here seems to be somewhat inaccurate. What we meant is that we have carefully selected data soueces based on mobile phone signaling data, combined with residents’ travel survey data , GIS data and navigation data to help identify the travel mode. The expression has been changed in line 528.
Reviewer 3 Report
Dear Authors,
I do not understand why you did not concern about the decision, "Reject", by a reviewer. It implies that your paper has serious problem in logical structure and soundness of the research approach. I do want to change some basic problems of "Random Forest Method or Random Decision Forest", because you just explained "learning algorithm composed of multiple decision trees". It is just filled with too much subjective articulations without any logical reasoning. The model should be explained its merits and disadvantages "compared with others based on the literature", not from your ideal arguments. You did not clarify the objective of the research, but the authors said to find the balancing way between demand and supply of transportation on "urban" area. Then, why did they choose "Kunshan, forest area"? There is no clear explanations on the logical background of the research design, objective selection, methodological appropriateness, and the precise description of the decision tree. There are too much logical missing links, but the authors just changed "few words" only. The analysis on the mountain or forest area does not improve the urban transportation, isn't it? I want to rewrite whole the paper based on the logical structure. For example, "Table 1. Data Source Introduction." shows the reason for the authors not to choose these. The title may mislead because it just seems the basic explantions of the research data. Morevoer, the contents of the table should show not just critic bad points, but the alternative bwtter approach of this paper. It seems just report, or manual for the new method of big data application.
Author Response
We sincerely thank you for the valuable comments on this paper, and your insightful suggestions are highly appreciated. First, the reason why choose RF compared with others has been explained in line 324-339. Second, about the question why choosing “Kunshan”, it is because: It is selected for the studies in this paper, mainly because the 4G telecommunication facilities are well established in its city area, which provides an ideal hardware/data condition for the topic addressed in this paper.(line 355-357). Third, Table 1 is a summary of different types of data sources according to the advatanges and limitaions not the explantions of the research data.We apologize that the previous version of paper mostly focus on the technical side of the addressed problem. As mentioned above, this travel mode identification accuracy is an important social issue, and we have added these discussions into the new version of the paper to improve its contributions.